# COVID-19 Pneumonia with Migratory Pattern in Agammaglobulinemic Patients: A Report of Two Cases and Review of Literature

Melania Degli Antoni [1,†], Verena Crosato [1,*,†], Francesca Pennati [1], Andrea Borghesi [2], Graziella Cristini [3], Roberto Allegri [3], Susanna Capone [3], Alberto Bergamasco [3], Annarosa Soresina [4], Raffaele Badolato [5], Roberto Maroldi [2], Eugenia Quiros-Roldan [1], Alberto Matteelli [1], Francesco Castelli [1] and Emanuele Focà [1]

1   Unit of Infectious and Tropical Diseases, Department of Clinical and Experimental Sciences, University of Brescia and ASST Spedali Civili di Brescia, 25123 Brescia, Italy
2   Department of Medical and Surgical Specialties, Radiological Sciences and Public Health, University of Brescia and ASST Spedali Civili of Brescia, 25123 Brescia, Italy
3   Unit of Infectious and Tropical Diseases, University of Brescia and ASST Spedali Civili di Brescia, 25123 Brescia, Italy
4   Pediatric Immunology Unit, Ospedale dei Bambini, ASST Spedali Civili di Brescia, 25123 Brescia, Italy
5   Molecular Medicine Institute "Angelo Nocivelli", Department of Clinical and Experimental Sciences, University of Brescia, 25123 Brescia, Italy
*   Correspondence: v.crosato001@unibs.it
†   These authors contributed equally to this work.

**Abstract:** X-linked agammaglobulinemia (XLA) is a primary immunodeficiency characterized by marked reduction in serum immunoglobulins and early-onset infections. Coronavirus Disease-2019 (COVID-19) pneumonia in immunocompromised patients presents clinical and radiological peculiarities which have not yet been completely understood. Very few cases of agammaglobulinemic patients with COVID-19 have been reported since the beginning of the pandemic in February 2020. We report two cases of migrant COVID-19 pneumonia in XLA patients.

**Keywords:** COVID-19; X-linked agammaglobulinemia; migrant pneumonia; organizing pneumonia; computed tomography

## 1. Introduction

Coronavirus disease 2019 (COVID-19) is an infectious disease caused by the severe acute respiratory syndrome coronavirus 2 (SARS-CoV-2) [1]. The typical radiological findings include bilateral and peripheral ground-glass opacities (GGOs) with or without patchy areas of consolidation [2–4]. The same disease in some groups of patients may present clinical and radiological peculiarities which have not yet been completely understood. In particular, immunosuppression can modulate the destructive effect of the immune system, slow down virus clearance, and change the expected course of the disease [5].

X-linked agammaglobulinemia (XLA) is a primary immunodeficiency characterized by marked reduction in serum immunoglobulins and by early-onset infections. The gene affected in XLA, Bruton tyrosine kinase (BTK), is located on the X-chromosome and its critical role in B cell development is evident by the universal B cell deficiency and absent precursor B cell differentiation in the bone marrow in patients with pathogenic mutations [6,7]. We report two cases of COVID-19 pneumonia with migratory pattern in XLA patients.

## 2. Case Descriptions

### 2.1. Case 1

On 1 April 2020, a 26-year-old patient affected by XLA was admitted to a tertiary hospital in northern Italy for fever, anorexia, and vomiting. The patient received intravenous

immunoglobulin injections every three weeks, but his general condition was good, and he was on no other chronical treatments. A chest X-ray performed on the day of admission showed bilateral GGOs involving the lower zones of both lungs. Nasopharyngeal swab tested positive for SARS-CoV-2.

Based on these findings, treatment for COVID-19 with dexamethasone (6 mg/day) and hydroxychloroquine (200 mg twice a day) was started. On April 8th, intravenous immunoglobulin administration (standard dose of 0.4 g/kg in single dose) was also administered. His baseline IgG value upon admission was 6.88 g/L (normal range 7–16 g/L). No oxygen supplement was needed but fever persisted over the following days, therefore antibiotic therapy with piperacillin/tazobactam was started on day 14 of hospitalization. Blood cultures were performed before the start of the antibiotic, but they were negative. Since he had been followed by our institutions for many years for his diagnosis of XLA,, he was transferred to our hospital on April 16th to continue his treatment for COVID-19. Treatment with dexamethasone and the antibiotic was continued and remdesivir was initiated, but it had to be discontinued due to liver toxicity. He also developed a skin rash which resolved spontaneously. No skin biopsy was performed. Fever and RT-PCR positivity on nasopharyngeal swab persisted, but no other antibiotic regimen was administered because of the lack of clinical, laboratory and radiological signs of bacterial superinfection. Furthermore, treatment with biological drugs was excluded because of the absence of a hyperinflammatory state.

A weekly chest computed tomography (CT) follow-up was performed during hospitalization and a peculiar radiological pattern was observed. On day 1 (Figure 1A), CT scan showed bilateral GGOs with small areas of consolidation in both lungs; on day 15 (Figure 1B), a migratory pattern of pulmonary opacities was observed, with a significant regression of GGOs in the lower lobes. Chest CT scans were also performed on day 22 and 28 (Figure 1C) and in both cases they showed a gradual regression of pulmonary opacities with appearance of new areas of both GGO and consolidation in different lung zones.

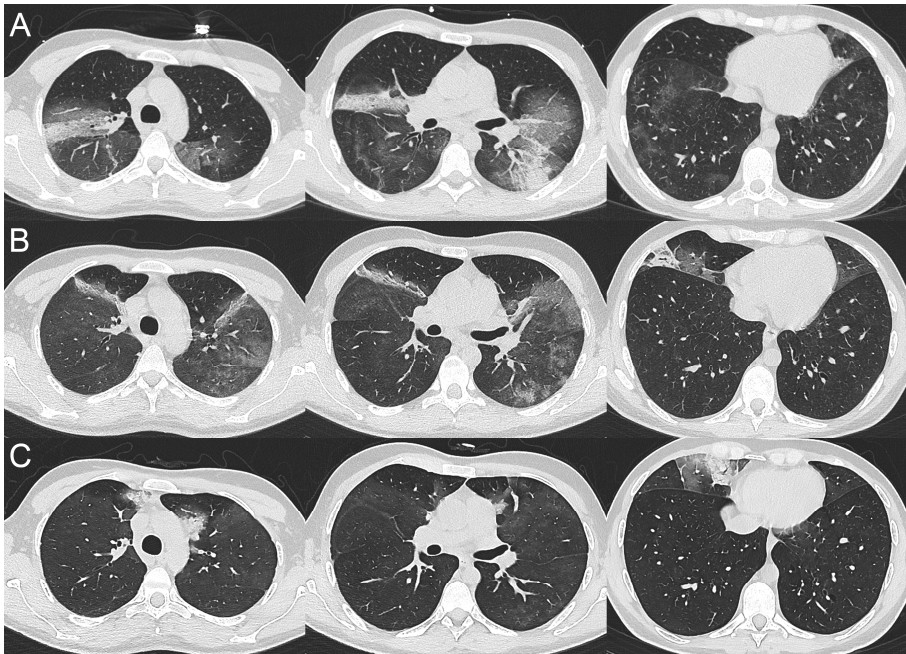

**Figure 1.** Series of axial computed tomography images with lung window setting, obtained at day 1 (**A**), day 15 (**B**) and day 28 (**C**) post-hospitalization, showing transient and migratory ground-glass opacities with small areas of consolidation in both lungs.

Subsequently, the radiological follow-up was continued with chest X-rays on day 35 and 41 and both examinations showed a progressive resolution of the pulmonary opacities in the lower zones with a persistence of consolidations in the upper zones. Respiratory gas

exchange was always preserved, without the need of oxygen supply. Moreover, the biochemical inflammation markers did not show a presence of hyperinflammation syndrome. The median C reactive protein (CRP) value during hospitalization was 17.6 mg/L (IQR 0.5–45.6 mg/L, normal range < 5 mg/L). A more frequent administration of intravenous immunoglobulins (in median every 7–10 days), to guarantee the maintenance of IgG levels in the normal range lead to the resolution of fever. The patient was discharged on day 42, apyretic, but with nasopharyngeal swab still positive for SARS-CoV-2. The only treatment continued after his discharge was intravenous immunoglobulin.

The negative nasopharyngeal swab was obtained on day 66, after more than 2 months.

*2.2. Case 2*

On 7 December 2020, a 41-year-old patient affected by XLA was admitted to a tertiary hospital in northern Italy for febrile symptoms but with no other manifestations. The patient suffered from chronic sinusitis and was in otherwise good clinical conditions. He worked as a butcher. His SARS-CoV-2 nasopharyngeal swab resulted negative. A chest X-ray, performed on the day of admission, showed pulmonary opacities in the upper zone of the right lung. Based on the patient's background and occupation, the main clinical suspicion was of mycobacterial pneumonia. A CT scan, performed at day 3 post-hospitalization, showed some GGOs with small areas of consolidation, interpreted as interstitial pneumonia (Figure 2A). On day 4, the patient underwent bronchoscopy with collection of bronchoalveolar lavage, which tested positive for SARS-CoV-2. Treatment with dexamethasone, enoxaparin and remdesivir was initiated, along with piperacillin/tazobactam and levofloxacin to prevent bacterial superinfection. No oxygen supply was needed. The inflammatory biochemical markers were not significantly increased. Ten days later, for a worsening of pulmonary gas exchange, a contrast-enhanced chest CT scan was performed. It excluded pulmonary embolism; however, a migratory pattern of pulmonary opacities was observed (Figure 2B).

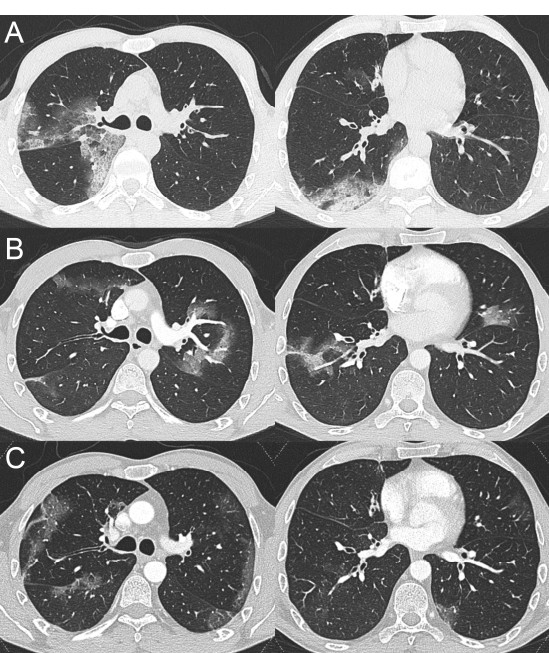

**Figure 2.** Series of axial computed tomography images with lung window setting, obtained at day 3 (**A**), day 16 (**B**), and day 44 (**C**) post-hospitalization, showing a similar migratory pattern of COVID-19 pneumonia.

On day 18, the patient developed a significant skin rash that was considered to be caused due to an allergic reaction to antibiotics. Piperacillin/tazobactam and levofloxacin were discontinued with no rash regression. Dexamethasone was continued and slowly

reduced. Control nasal swabs resulted negative. The patient's conditions during the following 20 days improved significantly, until day 43, when he suddenly developed high fever, desaturation and arthralgia. Therefore, another contrast-enhanced chest CT scan was performed to exclude pulmonary embolism and evaluate the trend of lung abnormalities. Again, no pulmonary embolism was observed; however, new peripheral GGOs were detected in both lungs (Figure 2C). A bronchoscopy was repeated, and SARS-CoV-2 RNA tested positive again. Following a rheumatologic examination on account of the skin rash and joint pain, the patient was diagnosed with concomitant Still-like disease, and corticosteroids and intravenous anakinra (100 mg/day) were initiated, to which the patient responded well, with regression of both the skin rash and the joint pain. On day 88, he was released from the hospital with oral prednisone and subcutaneous anakinra.

During his whole hospitalization period, the median CRP value was 28.8 mg/L (IQR 0.9–81.6 mg/L) and his IgG levels were kept in the normal range, thanks to regular immunoglobulin administration. However, at home he once more began experiencing high fever and desaturation, and on March 15th he was readmitted to the hospital. Nasopharyngeal swab and bronchoalveolar washing fluid examination highlighted the presence of SARS-CoV-2.

Anakinra dosage was doubled, and oral prednisone was switched to high dose intravenous dexamethasone (10 mg twice a day). Given the unique features of his COVID-19 disease, monoclonal antibody bamlanivimab was requested and obtained, and was administered on 22 March, with no adverse reactions. Nasopharyngeal swabs were performed on 23 and 26 March and resulted negative for SARS-CoV-2. The patient remained asymptomatic. On 29 March he was eventually discharged on anakinra and oral prednisone until further rheumatological indications.

## 3. Discussion

Italy was the first western country afflicted by COVID-19 pandemic and in particular Lombardy region was at the epicenter of the pandemic in Italy [8–10]. Since the outbreak of the SARS-CoV-2 pandemic in March 2020, many steps have been taken to comprehend this disease and its clinical and radiological manifestations. Chest CT has a potential role in the diagnosis, detection of complications and prediction of sequelae in COVID-19 patients. Several studies have been published reporting chest CT findings in COVID-19. The "typically" observed course is as follows: early stage COVID-19 often produces a GGO pattern with vascular enlargement located in the lower lobes with a posterior predominance. These findings progress to increased GGO and crazy paving pattern, then tend to consolidate, and the latter stage is characterized by a gradual decrease in the consolidation and GGOs, with eventual fibrosis onset [2,11–15].

Although this seems to be the most frequent presentation of COVID-19 pneumonia, a variety of other radiological manifestations has been described in the literature. The lack of a univocal pathognomonic pattern prompts us to investigate the clinical and radiological evolution of COVID-19 in different populations. Do immunocompromised patients with COVID-19 have different clinical and/or radiological presentations than the general population? [16].

Little is known about the spectrum of clinical presentations of COVID-19 in immunocompromised patients. Very few cases of COVID-19 in XLA patients have been reported in the literature to so far [17,18].

COVID-19 pneumonia in the majority of XLA patients, according to the literature, presented with overall mild symptoms, but rarely with respiratory failure requiring oxygen supplementation. Pharmacological BTK-inhibition has been reported to attenuate hyperinflammation in COVID-19, and dysfunctional BTK has been suggested as an explanation for mild disease in some patients with XLA. It can be hypothesized that the absence of humoral response protected from the immunopathology driven by SARS-CoV-2 specific antibodies with hyperinflammation [19,20]. In the literature, the most common reported

features include persistent fever, radiological evidence of bilateral interstitial pneumonia and delayed viral shedding, sometimes up to months after the first positive swab [21–27].

In a recent systematic review, Drzymalla et al., estimated a case fatality rate of 9%, hospitalization rate of 49%, and oxygen supplementation rate of 29% in people with primary immunodeficiency.

Overall, people with primary immunodeficiency, when infected, tested positive and showed symptoms for similar lengths of time as the general population. However, a number of patients with XLA were reported to have prolonged infections [28].

Here, we reported two cases of patients affected by XLA with prolonged SARS-CoV-2 infection and peculiar CT findings. Both patients presented with extensive bilateral GGOs which underwent complete resolution; however, follow-up CT showed new pulmonary opacities in different areas of both lungs without evidence of fibrotic changes.

If we take a closer look at their radiological findings, they appear compatible with the persistence of an initial pneumonia (GGO with or without small areas of consolidation), which however never seems to progress to more advanced stages of disease as observed in immunocompetent COVID-19 individuals (Figures 1 and 2). Another interesting finding in both cases is the migratory pattern of pulmonary opacities (Figures 1 and 2).

These radiological features are compatible with the presence of an initial alteration of the lung parenchyma without further progression to more advanced stages of disease, as it would be expected during COVID-19 pneumonia. Why does this happen? A probable hypothesis could be, it is because of the lack of immunomediated response in patients affected by XLA, which impairs the establishment of a proper inflammatory reaction.

On the other hand, we believe that COVID-19 should be included in the differential diagnosis of migratory pulmonary opacities along with cryptogenic organizing pneumonia (OP) and secondary OP (associated with a large variety of predisposing conditions, such as immunological diseases and lymphoproliferative disorders). OP is a sub-acute process of pulmonary tissue repair secondary to lung injury, defined histopathologically by intra-alveolar buds of granulation tissue within the lumen of distal pulmonary airspaces. It can be either cryptogenic or secondary (SOP) to different clinical conditions, namely infections. OP secondary to viral infections are usually suspected when the patient, after an initial improvement, clinically worsens with reappearance of fever, increasing inflammatory markers and damaged oxygen exchange and respiratory mechanism [29–31].

Very few cases of OP in patients with COVID-19 were described [32–34]. In particular, two cases of OP were observed in hematologic patients, where chest CT scans showed migratory pulmonary opacities without signs of pulmonary thromboembolism. A presumptive diagnosis of OP was made on the basis of prolonged clinical progression and typical CT findings [35,36].

OP, confirmed radiologically and histologically, as an active and sometimes aberrant lung repair process, may represent the evolution of COVID-19 in patients with immunological disorders, with mild to moderate disease and no development of hyperinflammatory state.

In conclusion, primary immunological defects determine responses to external insults, such as viral infections, that cannot always be predicted. Though very few cases of agammaglobulinemic patients with SARS-CoV-2 have been reported since the beginning of the pandemic in February 2020, the particular findings presented here suggest that migrant pneumonia could be a further radiological pattern of COVID-19, typical, but not necessarily exclusive, in immunosuppressed patients. Further studies are required to determine whether these findings could be considered recurrent radiological manifestations of SARS-CoV-2 infection in primary immunodeficient patients. Finally, clinicians involved in the COVID-19 fight need to be aware that migrant pneumonia is a possible new clinical and radiological manifestation of this disease.

**Author Contributions:** Conceptualization: M.D.A., E.F., A.B. (Andrea Borghesi); Original Draft Preparation: M.D.A., V.C., F.P., E.F., A.B. (Andrea Borghesi), Methodology, Validation, Investigation, Resources, Data Curation, Review and Editing, Visualization, Supervision: M.D.A., E.F., A.B. (Andrea

Borghesi), V.C., F.P., G.C., R.A., S.C., A.B. (Alberto Bergamasco), A.S., R.B., R.M., E.Q.-R., A.M., F.C. All authors have read and agreed to the published version of the manuscript.

**Funding:** This research received no external funding.

**Institutional Review Board Statement:** Not applicable.

**Informed Consent Statement:** Written informed consent has been obtained from the patient to publish this paper.

**Data Availability Statement:** Not applicable.

**Acknowledgments:** This work was accepted as poster at ECCMID 2021.

**Conflicts of Interest:** The authors declare no conflict of interest.

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
