# Peer review of "COVID-19 Pneumonia with Migratory Pattern in Agammaglobulinemic Patients: A Report of Two Cases and Review of Literature"

_tomography, doi:10.3390/tomography9030073_

Round 1

Reviewer 1 Report

This is an interesting and very well-written case series and literature review about migrant COVID-19 pneumonia in X-linked agammaglobulinemic patients. However, there are some issues that should be addressed by the authors:

- Title: The title is currently self-repeating: COVID-19 pneumonia with migratory pattern in 2 agammaglobulinemic patients: a report of two cases and review of literature-19 pneumonia with migratory pattern in agammaglobulinemic patients: a report of two cases and review of literature

- Case 1 Description: Page 2 Lines 64 and 79: (administered?) (preserved?)

- Discussion: Please avoid repetition of definitions (e.g. Page 4 Line 158. XLA was defined in the previous paragraphs, Organizing pneumonia is defined as OP at line 175, but in the immediately following sentence the term organizing pneumonia is again used unabbreviated).

Abbreviations are not necessary for words that are only used once (e.g. SOP).

Page 4 from line 168 to line 172, is unclear. Please, rephrase.

From Line 190 to line 193, is unclear. Please, rephrase.

Both cases presented skin rash, in the authors' opinion, is there a pathogenetic link? Or did the two patients get skin rash for different reasons? Please explain.   

Author Response

This is an interesting and very well-written case series and literature review about migrant COVID-19 pneumonia in X-linked agammaglobulinemic patients.

We thank the reviewer for the positive comment and appreciation.

However, there are some issues that should be addressed by the authors:

- Title: The title is currently self-repeating: COVID-19 pneumonia with migratory pattern in 2 agammaglobulinemic patients: a report of two cases and review of literature-19 pneumonia with migratory pattern in agammaglobulinemic patients: a report of two cases and review of literature.

We thank you for your advice. The title was corrected.

- Case 1 Description: Page 2 Lines 64 and 79: (administered?) (preserved?).

We thank you for this valuable suggestion. We have used the “administered” term.

- Discussion: Please avoid repetition of definitions (e.g. Page 4 Line 158. XLA was defined in the previous paragraphs, Organizing pneumonia is defined as OP at line 175, but in the immediately following sentence the term organizing pneumonia is again used unabbreviated). Abbreviations are not necessary for words that are only used once (e.g. SOP).

We modified the text as requested by eliminating the full words and leaving only the abbreviations.

- Page 4 from line 168 to line 172, is unclear. Please, rephrase.

We rephrased the sentence to make it clearer.

- From Line 190 to line 193, is unclear. Please, rephrase.

We reformulated the sentence to make it clearer.

- Both cases presented skin rash, in the authors' opinion, is there a pathogenetic link? Or did the two patients get skin rash for different reasons? Please explain.

We thank the reviewer for this question.  We don’t believe there is a pathogenetic link between the skin rash and the COVID-19 related manifestations in our two patients. We couldn’t find an explanation for the skin rash in our first case, but it resolved spontaneously shortly after its appearance, and was therefore not further investigated. For what concerns our second case, it was first interpreted as a possible allergic reaction, but further identified as Still-like disease. Therefore, he was initiated on anakinra with good response.

We decided to leave the skin rash out of our discussion as it doesn’t seem relatable to the migrant lung pattern during COVID-19 in XLA.

Reviewer 2 Report

The authors presented the clinical course of COVID infection in 2 patients with Bruton's agammaglobulinaemia. As Bruton's agammaglobulinaemia is a rare condition, the paper is worthy of publication. However, I have numerous concerns with the current version of the article.

Overall, the presentation of the cases is rather chaotic and therefore difficult to analyse. I suggest presenting the course of the disease and treatment in the form of a diagram. There is no information on the indications for the treatment given, especially as it was different in both cases. The doses of medication and the dose of immunoglobulin used are missing - substitution or immunosuppression? what dose per kilogram of body weight?  There are no results of laboratory tests (at least CRP, morphology). There is no information regarding IgG levels at the time of the illness. Was proper IgG level control maintained? It is very important.

Is anything known about the lungs before COVID? Did the patients have any lesions?

What is missing from the discussion is a comparison with other described cases of Bruton with COVID. It is also worth comparing this course with other patients with immunodeficiency - e.g. CVID.

Numerous linguistic and stylistic errors - linguistic revision recommended

Case 1:

What was the reason for the delay in starting the anti-viral medication?

“more frequent administration of intravenous immunoglobulins” - please clarify

What were the indications for such a frequent CT scan of the chest?

Case 2

“On December 11th, the patient underwent bronchoscopy with collection of bronchoalveolar washing fluid, which was tested positive for SARS-CoV-2”  – And the result of the nose test? Because there is no information.

Author Response

The authors presented the clinical course of COVID infection in 2 patients with Bruton's agammaglobulinaemia. As Bruton's agammaglobulinaemia is a rare condition, the paper is worthy of publication. However, I have numerous concerns with the current version of the article.

Overall, the presentation of the cases is rather chaotic and therefore difficult to analyse.

We revised the text and modified it to make it clearer and more comprehensible to the reader.

- I suggest presenting the course of the disease and treatment in the form of a diagram.

Considering the differences in management and time frame of the two patients, we are afraid the two diagrams wouldn’t be comparable. However, we widened the description of the cases to make them more comprehensible and complete.

- There is no information on the indications for the treatment given, especially as it was different in both cases. The doses of medication and the dose of immunoglobulin used are missing - substitution or immunosuppression? what dose per kilogram of body weight?

Thank the reviewer for the comment. Normally, immunoglobulins are administered at a dose of 0,4 g/Kg once every two weeks, with changes in the frequency of administration according to the blood levels of IgG. We added immunoglobulin dosage and frequency of administration in the text.

- There are no results of laboratory tests (at least CRP, morphology). There is no information regarding IgG levels at the time of the illness. Was proper IgG level control maintained? It is very important.

Thank you for your comment. CRP and IgG values are available for both patients and we now inserted them in the text. For both patients no significant abnormalities in the blood cell count and leukocyte morphology were observed, and we therefore decided to leave this data out of the description, as we considered it not significant for the purpose of this paper.

The patients were administered immunoglobulins periodically to assure proper maintenance of IgG levels throughout the whole hospital stay, and after discharge.

- Is anything known about the lungs before COVID? Did the patients have any lesions?

Thank you for this question. None of them had previous pulmonary lesions despite their long history of XLA.

- What is missing from the discussion is a comparison with other described cases of Bruton with COVID. It is also worth comparing this course with other patients with immunodeficiency - e.g. CVID.

We understand the concern, and updated our references list, by adding other clinical cases and systematic reviews on the topic. We would like to specify that the objective of our report is an accurate description of a specific radiologic pattern we observed in XLA patients during COVID-19. Even though the clinical features of COVID-19 in XLA are similar to those observed in other primary immunodeficiencies (e.g. CVID), we decided not to compare our cases with other pathologies, as we didn’t find the same radiological alterations and migratory pattern in patients affected by immunodeficiencies other than XLA.

- Numerous linguistic and stylistic errors - linguistic revision recommended. Case 1: What was the reason for the delay in starting the anti-viral medication? “More frequent administration of intravenous immunoglobulins” - please clarify. What were the indications for such a frequent CT scan of the chest?

We thank the reviewer for the comment and has a language revision performed.

Concerning Case 1, antiviral treatment was delayed because the patient was initially admitted at another hospital, and subsequently transferred to our unit. As specified above, immunoglobulins are administered with a frequency of one dose every two weeks. We added the specific dosage to the text.

As for what concerns the frequency of the CT scans, given the persistent positivity to SARS-CoV-2 and the patient’s immunological condition, regular CT scans were performed as indicated by our immunology consultant.

- Case 2 “On December 11th, the patient underwent bronchoscopy with collection of bronchoalveolar washing fluid, which was tested positive for SARS-CoV-2” – And the result of the nose test? Because there is no information.

We thank for the comment. The patient was tested with a swab on the day of admission, and it resulted negative. Given the strong clinical suspicion, he therefore underwent a bronchoscopy with subsequent finding of SARS-CoV-2 positivity.

We added the missing information in the text.

Round 2

Reviewer 2 Report

The authors addressed the more important points in my previous review. I accept the comments. I still believe that the doses of the drugs used should be stated. Additional comments below:

"treatment for COVID-19 with dexamethasone and hydroxychloroquine was started" >>> doses

"his general conditions were good" >>> it should be" his general condition was good 

Case 1 >>> IgG level on admission

case 2: "Anakinra dosage was doubled, and oral prednisone was switched to high dose intravenous dexamethasone" >>> doses.

discussion:

"people with primary immunodeficiency, when infected, tested positive and
showed symptoms for similar lengths of time as the general population. However, a number of people with... >>>> perhaps use " patients" instead of repeating the word "people"

"with mild to moderate disease and no development of hyperinflammatory state. . >>> double dot
